# Distilled Cross-Combination Transformer for Image Captioning with Dual Refined Visual Features

## ABSTRACT

Transformer-based encoders that encode both region and grid features are the preferred choice for the image captioning task due to their multi-head self-attention mechanism. This mechanism ensures superior capture of relationships and contextual information between various regions in an image. However, because of the Transformer block stacking, self-attention computes the visual features several times, increasing computing costs and producing a great deal of redundant feature calculation. In this paper, we propose a novel Distilled Cross-Combination Transformer (DCCT) network. Specifically, we first design a distillation cascade fusion encoder(DCFE) to filter out redundant features in visual features that affect attentional focus, obtaining refined features. Additionally, we introduce a parallel cross-fusion attention module (PCFA) that fully utilizes the complementarity and correlation between grid and region features to better fuse the encoded dual visual features. Extensive experiments on the MSCOCO dataset demonstrate that the proposed DCCT strategy outperforms many state-of-the-art techniques and attains exceptional performance.

## CCS CONCEPTS

• **Computing methodologies** → **Natural language generation**; *Computer vision tasks*;

## KEYWORDS

Image captioning; Cross combination; Contrastive Language-Image Pre-training; Reinforcement learning

## 1 INTRODUCTION

Image captioning is an interdisciplinary research field at the intersection of computer vision and natural language processing. Its goal is to automatically understand the content of an image and generate natural language descriptions closely related to it. It is a challenging task that involves analyzing cross-modal data. With the significant success of region-based features in image captioning tasks[2], many researchers have continued to improve the performance of image caption generation based on these features[6, 18, 27]. Despite their tremendous contribution as the sole visual feature in image captioning, critiqued for their lack of fine-grained details and contextual information, these methods stand in contrast to grid features[23, 25], which retain all the information within a given image.

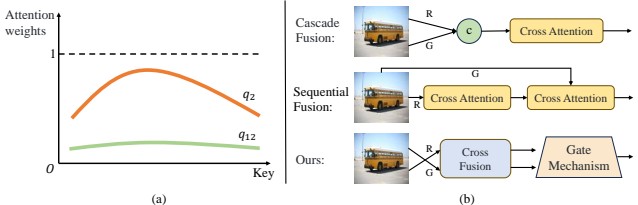

**Figure 1: Left (a): Line chart of attention weights for two randomly selected query vectors $q$ in image features against all key values. Right (b): Two common fusion methods for the decoder-side cross-attention module using dual visual features, and our proposed parallel cross-fusion method.**

Incorporating the triumph of Transformers in natural language processing[35], numerous NLP endeavors have leveraged its distinctive encoder-decoder structure, a trend that has garnered significant interest within the realm of image captioning. Transformer-based image captioning models[6, 8, 41, 42, 45] have excellent modeling capabilities, employing multi-head attention mechanisms to simultaneously attend to different parts of both the image and text while passing global contextual information between the encoder and decoder. Having demonstrated exceptional performance, Transformers has become the dominant approach to captioning over the past few years.

However, during the stacking process of Transformer blocks, multiple self-attentions are performed on all region features and grid features. This may result in the inclusion of many irrelevant features, affecting the attention focus on important features. Represented in (a) of Figure 1, the key values of visual features post-linear projection are depicted on the horizontal axis, while the vertical axis illustrates the attention weights. The curvature of the curve indicates the level of activation for the corresponding q-value. The more curved the curve, the more active the q-value, indicating the higher value of the corresponding feature. Conversely, a less curved curve indicates a lower value of the corresponding feature. For $q_{12}$, it has similar and low attention weights with all key values, indicating that it does not correlate with most features. On the other hand, $q_2$ is different; it has higher attention weights with some key values, demonstrating its correlation with many features. Therefore, inspired by the time-series prediction model Informer[52], we attempt to replace the original Transformer encoder with an improved Informer encoder.

In addition, to fully utilize the region and grid features obtained from the encoding layer during the decoding stage, they are fused in the cross-modal attention module. There are various methods for designing cross-modal attention in the middle part. Some common methods include sequential cross-modal attention and cascaded cross-modal attention. As shown in (b) in Figure 1, in cascaded cross-modal attention, the grid and region features obtained from the

encoding stage are concatenated and then fed into the cross-modal attention module. Sequential cross-modal attention calculates two independent cross-modal attention for region features and grid features separately and then combines them. Although sequential cross-modal attention can independently process region features and grid features through sequential design, it still has two main problems: 1) The former may not fully exploit the interaction between different features, thereby limiting the model's expressive power and performance. 2) The order of the two cross-modal attention in the latter may affect the final performance. If the region features are computed first, there may be insufficient attention on the grid features, and vice versa.

To address the aforementioned issues, we present a unique Distilled Cross-Combination Transformer (DCCT) to enhance the efficiency and accuracy of image captioning. Firstly, we put forward a Distillation Cascade Fusion Encoder (DCFE), which filters out redundant visual features like $q_{12}$ to make attention focus more on important features like $q_2$, resulting in more refined feature representations. Secondly, we propose a Parallel Cross-modal Fusion Attention (PCFA) module, which cross-attends to processed grid features and region features, fully utilizing the correlation and complementarity between them. Then, a gating mechanism is applied to adjust the importance and contribution of the two multimodal features to caption generation, enabling the model to better utilize the information from different features and achieve comprehensive multimodal fusion. The contributions made in this article can be summarised in the following way:

- We present a Distillation Cascade Fusion Encoder (DCFE), which enhances encoding efficiency by filtering out redundant features from the images to produce more refined visual representations.
- We introduce a novel Parallel Cross-modal Fusion Attention (PCFA) module that fully exploits the complementarity and correlation between dual visual data to obtain more informative multimodal feature representations.
- Extensive experiments on the benchmark MS COCO dataset show that our suggested DCCT outperforms the most advanced methods, achieving an exceptional performance of 144.1 in the ensemble configuration.

## 2 RELATED WORK

**Image Captioning.** In the early days, research on image captioning focused mainly on template-based and retrieval-based methods[9, 26, 34, 37]. With the rise of deep learning techniques, particularly the outstanding performance of convolutional neural networks (CNNs) in image feature extraction and the successful application of recurrent neural networks (RNNs) in sequence modeling, the image captioning task has ushered in new development opportunities. The model proposed by Vinyals et al.[39] adopts a CNN-RNN encoder-decoder architecture to achieve end-to-end mapping from images to text. The introduction of attention mechanisms further improves the accuracy and detail of image captioning. Anderson et al.[2] proposed a bottom-up and top-down attention mechanism that extracts features from different regions of the image and focuses attention on the most informative region. Subsequently, with the significant success of Transformer models in natural language

processing, many researchers began to introduce different variants of Transformers in image captioning tasks to continuously improve performance. For example, the M2 model proposed by Cornia et al.[6] prior knowledge through a mesh-like connection between memory vectors and encoding-decoding modules. The DLCT model proposed by Luo et al.[23] achieves complementary features of region and grid in image captioning. Recently, large-scale data pre-trained visual-language models (VLMs)[10, 31, 33] have achieved great success in various visual-language tasks, such as video description generation[33, 33, 36], visual question answering[7, 14, 15], and image captioning[10, 22, 46]. Visual features extracted by different large models have unique advantages and powerful representation capabilities. For example, in VinVL[50], the authors use stronger object detectors to extract diverse region features, providing compact object-level representations in images. In CLIP[31], the authors obtained cross-modal grid features that contain rich semantic scenes and fine-grained details through contrastive pre-training. These advanced visual features are used for image captioning tasks, helping the model to perform complex reasoning and improve the accuracy of generated descriptions.

**Visual-Semantic Interaction** In the task of image captioning, the interaction between vision and language during the decoding stage is a crucial step[6, 16, 19, 49]. Lu et al.[21] introduced the concept of an additional visual sentinel and extended the previous LSTM architecture. This adaptive mechanism allows the model to determine whether to focus attention on the language or visual part during the decoding process, thereby enhancing the interaction between vision and language. Chen et al.[4] aimed to bridge the semantic gap between different modalities. They designed a novel encoder-decoder attention mechanism with an unsaturated calibration gate function to control the interaction between vision and language. This mechanism helps to achieve a balance between the two modalities. Li et al.[17] utilized CLIP to extract grid features and employed cross-modal retrieval to identify essential semantic clues. They incorporated cross-attention in the decoder to facilitate the interaction between vision and language, enabling modality fusion. Zhang et al.[48] recognized that the semantic information obtained from offline detectors often contains irrelevant objects. They proposed a novel constrained weakly supervised learning module, which provides more relevant semantic-enhanced information to improve the model's visual-semantic interaction capability.

In general, although the above-mentioned methods have partially alleviated the semantic gap and achieved basic visual-semantic interaction, they still fail to fully integrate the semantic information in visual content with textual information. In this paper, we propose a novel Distilled Cross-Combination Transformer (DCCT) for image caption generation. Detailed explanations will be provided in Section 3.

## 3 METHODLGY

This article introduces a novel Distilled Cross-Combination Transformer (DCCT), as shown in the overall framework diagram in Figure 3. During the encoding stage, given an image, we first extract grid features using the widely adopted pre-trained model CLIP[31], and region features using the object detector from the pre-trained model VinVL[50]. To unify the feature dimensions, we

project these features onto a specified dimension $d$, indicated by $V_I^{L_G} = \{v_i\}^{L_G}$ and $V_I^{L_R} = \{v_i\}^{L_R}$ individually. Here, $L_G$ and $L_R$ represent grid and region feature numbers, respectively. Afterward, we input them into the DCFE for encoding. After passing through N self-attention distillation modules and a cascading layer, we obtain the grid output features $V_O^{N_G}$ and region output features $V_O^{N_R}$, defined as follows:

$$(V_O^{N_R}, V_O^{N_G}) = DCFE(V_I^{L_R}, V_I^{L_G}). \quad (1)$$

During the decoding stage, these refined grid features and region features are fed into a Parallel Cross-Fusion Attention (PCFA) module to be combined with the word features $w_t^{i+1}$ through a multi-head masked self-attention layer, resulting in the fused feature $p_t^{i+1}$. The corresponding definition is as follows:

$$p_t^{i+1} = PCFA(w_t^{i+1}, V_O^{N_G}, V_O^{N_R}). \quad (2)$$

The details of PCFA will be described in Section 3.2. Afterward, we pass rich multimodal representation $p_t^{i+1}$ through a position-wise feed-forward layer, followed by residual connections and layer normalization, to obtain the output $y_t^{i+1}$. The corresponding definition is as follows:

$$y_t^{i+1} = LayerNorm(p_t^{i+1} + FFN(p_t^{i+1})). \quad (3)$$

Finally, the output of the $N$-th layer is fed into a vocabulary classifier for predicting the next word.

Next, we will explain the two most important modules in DCCT: the Distillation Cascade Fusion Encoder and the Parallel Cross-Fusion Attention module.

## 3.1 Distillation Cascade Fusion Encoder

In this paper, we propose a Distillation Cascade Fusion Encoder (DCFE), which consists of N multi-head probability sparse attention layers and a cascade layer. The probability sparse attention layer includes probability sparse attention, position-wise feed-forward layer, and convolutional pooling layer. The two primary sections of this encoder component are: multi-head probability sparse self-attention and encoding visual features using DCFE. We will provide a detailed introduction to these in the following.

**Multi-head ProbSparse Self-Attention.** Regarding the ProbSparse Self-Attention, we will use grid features $V_I^{L_R}$ to illustrate since the encoding process for grid features and region features is similar. We represent the visual features as $Q$, $K$, and $V$. Here, the visual features are vector representations obtained through linear transformations of grid features $V_I^{L_R}$. In an image, not every position's attention is necessarily important. For each query $Q$, only a small portion of $K$ has a strong relationship with it. Calculating every $Q$ with every $K$ would be inefficient. As shown in Figure 2 (a), for the attention weight matrix heat map of grid features, it can be observed that only a small portion of $Q$ and $K$ have relatively large attention weights, appearing brighter in the image, while the majority of attention weights are small or even zero, shown as black in the image. Similarly, the region feature heat map in Figure 2 (b), follows a similar pattern to the grid features, which will not be reiterated here.

Specifically, in the first step, we sample the queries $Q$ to determine which ones are more correlated with each other and which ones have lower correlations. We begin by randomly sampling $K$

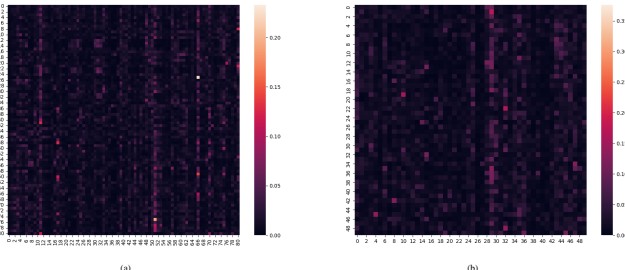

(a)
(b)

**Figure 2: Left (a): Heat map of the attention weight matrix for the grid features of any image, where brighter regions indicate higher weights. Right (b): Heat map of the attention weight matrix for the region features of any image, similar to the grid features, with brighter regions corresponding to higher weight coefficients.**

to obtain $r\_k$. The number of sampled $K$ is given by $c \cdot \lfloor \log_{M_K} \rfloor$, where $c$ is a constant and $M_K$ represents the number of keys $K$. Next, we calculate the similarity between $Q$ and the sampled $r\_k$ as $q_m k_n^\top / \sqrt{d}$. To improve computational efficiency, we measure the difference between the maximum value among these similarities and a uniform distribution. A larger difference indicates a more significant relationship. The corresponding definitions are as follows:

$$\bar{S}(q_m, K) = \max_n\{\frac{q_m k_n^\top}{\sqrt{d}}\} - \frac{1}{L_K}\sum_{n=1}^{L_K}\frac{q_m k_n^\top}{\sqrt{d}}. \quad (4)$$

Next, we sort the calculated similarity differences $\bar{S}(q_m, K)$ in descending order and select the top $c \cdot \lfloor \log_{M_Q} \rfloor$ features $Q$. We then compute attention between these selected features $Q$ and all $K$ and $V$ vectors. The corresponding definitions are as follows:

$$Q_{selected} = Attention(\tilde{Q}, K, V) = Softmax(\frac{\tilde{Q}K^\top}{\sqrt{d}})V, \quad (5)$$

where $\tilde{Q}$ represents the selected features $Q$ after sorting.

For the remaining features $Q$, since their correlations with other features $Q$ are relatively low and their activity levels are lower, we do not compute their attention. Instead, we replace their similarity relationships with other features $Q$ by using the mean of all $V$ vectors. Afterward, we fill the selected $Q_{selected}$ features based on their indices into the original positions in $V$, replacing the original mean vectors. Then, we apply residual connections and normalization to obtain the output visual features. We thus finish the Multi-head ProbSparse Self-Attention process on the original visual features. The corresponding definitions are as follows:

$$V_{mean} = mean(V), \quad (6)$$
$$V_{fill} = fill(V_{mean}, Q_{selected}), \quad (7)$$
$$\tilde{V} = LayerNorm(V + V_{fill}), \quad (8)$$

where the mean function is used to calculate the mean vector, the fill function represents filling, and LayerNorm represents layer normalization.

**The process of encoding visual features with DCFE.** For the entire encoding process, specifically, given the initial grid features

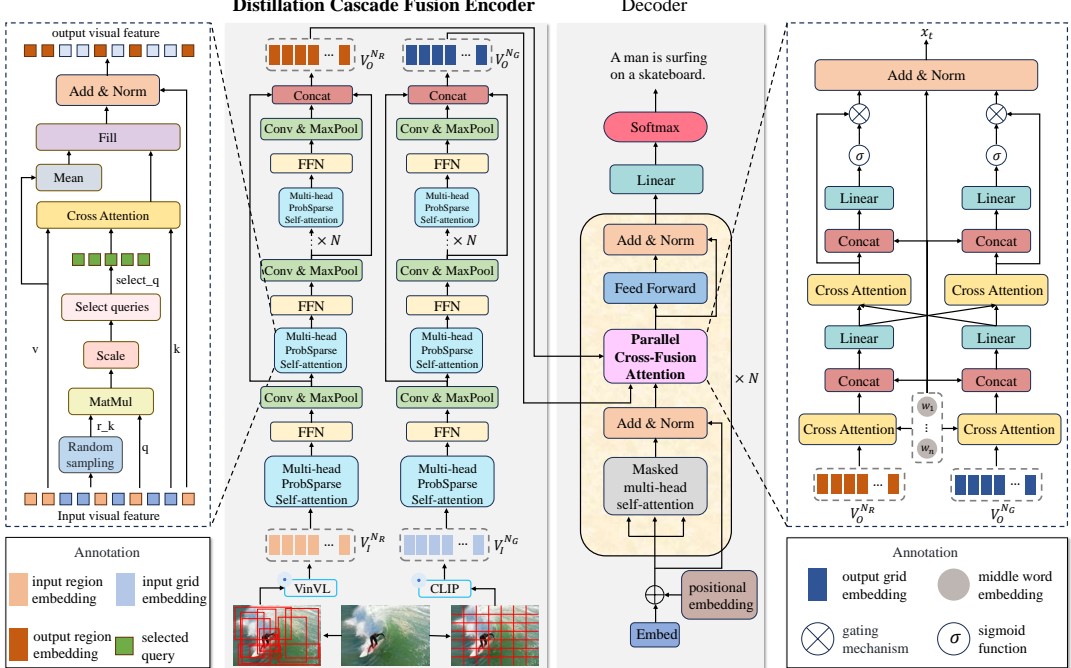

**Figure 3: Overview of the architecture of DCCT. Given an input image, CLIP[31] and VinVL[50] extract grid features and region features respectively. Subsequently, the dual visual features are fed into the DCFE for encoding to obtain refined visual features. Next, through the PCFA module in the decoder, the encoded dual comprehensive visual features are cross-fused with textual features. Finally, the multimodal features are passed to the next decoding layers for additional processing.**

$V_j^{M_G} = \{v_i\}_{i=1}^{M_G}$ and the initial region features $V_j^{M_R} = \{v_i\}_{i=1}^{M_R}$ of the $(j+1)$-th layer of the multi-head ProbSparse self-attention, where $M_G$ and $M_R$ represent the number of initial visual features in the $j$-th layer. First, the grid features $V_j^{M_G}$ and region features $V_j^{M_R}$ are separately fed into the multi-head ProbSparse self-attention for calculation. The corresponding definitions are as follows:

$$\tilde{G}_{j+1} = MPS(V_j^{M_G}, V_j^{M_G}, V_j^{M_G}), \tag{9}$$

$$\tilde{R}_{j+1} = MPS(V_j^{M_R}, V_j^{M_R}, V_j^{M_R}), \tag{10}$$

where MPS represents Multi-head ProbSparse Self-Attention.

Then we pass it through a position-wise feed-forward layer, followed by residual connections and layer normalization, to obtain the outputs $\bar{G}_{j+1}$ and $\bar{R}_{j+1}$. The corresponding definitions are as follows:

$$\bar{G}_{j+1} = LayerNorm(\tilde{G}_{j+1} + FFN(\tilde{G}_{j+1})), \tag{11}$$

$$\bar{R}_{j+1} = LayerNorm(\tilde{R}_{j+1} + FFN(\tilde{R}_{j+1})). \tag{12}$$

We keep stacking layers after one cycle of position-wise feed-forward layer and ProbSparse self-attention. However, unlike traditional transformers, due to the presence of redundant combinations of $V$ in the visual feature map of the encoder, we perform a distillation operation on these visual features obtained through probabilistic sparse attention. This operation prioritizes selecting dominant and active features and forms a refined self-attention feature map in the next layer. The distillation process is carried out from the $j$-th layer to the $(j+1)$-th layer through the following steps:

$$G_{j+1} = MaxPool(ELU(BN(Conv1d(\bar{G}_{j+1})))), \tag{13}$$

$$R_{j+1} = MaxPool(ELU(BN(Conv1d(\bar{R}_{j+1})))), \tag{14}$$

where $Conv1d(\cdot)$ represents the convolution operation, which performs a 1-D convolution filter (kernel width = 3) on the feature dimension. BN stands for batch normalization, ELU represents the ReLU activation function, and MaxPool indicates maximum pooling.

Finally, we concatenate the visual features obtained from each layer of ProbSparse self-attention to obtain the final visual features $V_O^{N_G} = [G_i|_{i=1}^{N_G}]$ and $V_O^{N_R} = [R_i|_{i=1}^{N_R}]$. As a result, we can combine feature data from several levels and produce a more complete feature representation.

## 3.2 Parallel Cross-Fusion Attention

The cross-modal parallel cross-fusion attention module we propose mainly consists of three inputs: the grid features $V_O^{N_G}$ encoded by the encoder, the region features $V_O^{N_R}$ encoded by the encoder, and the hidden state $w_t^{(i+1)}$ obtained from the decoder's masked self-attention sub-layer, where we describe the $(i+1)$ block of the decoder. First, the grid features $V_O^{N_G}$ obtained from the encoder are fed into the cross-attention module as keys and values. Combined with the hidden state $w_t^{(i+1)}$, a cross-attention mechanism is applied to capture the relationship between word features and region

features. Combined with residual connections and layer normalization, the resulting attended features yield the multimodal features $\bar{g}_t^{i+1}$. Similarly, the modeling of cross-attention between grid features and word characteristics is similar to that of region features. After the same module calculation, the multimodal feature $\bar{r}_t^{i+1}$ is obtained. Therefore, their respective operations are as follows:

$$g_t^{i+1} = CA(w_t^{i+1}, V_O^{N_G}, V_O^{N_G}), \tag{15}$$

$$\bar{g}_t^{i+1} = LayerNorm(g_t^{i+1} + w_t^{i+1}), \tag{16}$$

$$r_t^{i+1} = CA(w_t^{i+1}, V_O^{N_R}, V_O^{N_R}), \tag{17}$$

$$\bar{r}_t^{i+1} = LayerNorm(r_t^{i+1} + w_t^{i+1}), \tag{18}$$

where CA represents Cross-Attention and LayerNorm represents Layer Normalization.

We concatenate the multimodal features $\bar{g}_t^{i+1}$ and $\bar{r}_t^{i+1}$ with the word features $w_t^{i+1}$, and then use a linear transformation to convert into $d$-dimensional vectors, obtaining the integrated features $\tilde{g}_t^{i+1}$ and $\tilde{r}_t^{i+1}$. The purpose of this is to further integrate the information of word characteristics, grid characteristics, and region characteristics, and transform the integrated information features into a specific dimension for subsequent modules to learn and compute, to generate more comprehensive and rich representations. The corresponding definitions are as follows:

$$\tilde{g}_t^{i+1} = W^g[\bar{g}_t^{i+1}; w_t^{i+1}] + b^g, \tag{19}$$

$$\tilde{r}_t^{i+1} = W^r[\bar{r}_t^{i+1}; w_t^{i+1}] + b^r, \tag{20}$$

where [.] denotes concatenation, $W^g$ and $W^r$ represent the corresponding weight matrices, and $b^g$ and $b^r$ represent the bias terms.

We cross-fuse the integrated dual visual features, which helps the model better understand and utilize both image and text information at a macro level, reducing dependence on individual features. In addition, compared to cascading fusion and sequential fusion, this cross-fusion approach can explore deeper-level visual and semantic connections, effectively compensating for the limitations of region features and enriching the representation of multimodal features, resulting in more accurate descriptions. Specifically, for the grid feature branch, we use the integrated region feature $\tilde{r}_t^{i+1}$ as keys and values, and the integrated grid feature $\tilde{g}_t^{i+1}$ as queries. We then compute the cross-attention to thoroughly explore their correlations. The resulting cross-attention features are then passed through residual connections and layer normalization to obtain enriched multimodal features $g_t^{R(i+1)}$. Similarly, for the region feature branch, the learning process is similar to the grid feature branch, where after a series of computations, the feature $r_t^{G(i+1)}$ is obtained.

$$g_t^{'(i+1)} = CA(\tilde{g}_t^{i+1}, \tilde{r}_t^{i+1}, \tilde{r}_t^{i+1}), \tag{21}$$

$$g_t^{R(i+1)} = LayerNorm(g_t^{'(i+1)} + \tilde{g}_t^{i+1}), \tag{22}$$

$$r_t^{'(i+1)} = CA(\tilde{r}_t^{i+1}, \tilde{g}_t^{i+1}, \tilde{g}_t^{i+1}), \tag{23}$$

$$r_t^{G(i+1)} = LayerNorm(r_t^{'(i+1)} + \tilde{r}_t^{i+1}). \tag{24}$$

Finally, we concatenate these enriched multimodal features $g_t^{R(i+1)}$ and $r_t^{G(i+1)}$ with the initial word feature $w_t^{i+1}$ and then project them into a $d$-dimensional vector through a linear transformation. At the same time, we use the sigmoid function to normalize them into

weighting factors $\alpha_t^{g(i+1)}$ and $\alpha_t^{r(i+1)}$ respectively. Subsequently, we multiply the multimodal features $g_t^{R(i+1)}$ and $r_t^{R(i+1)}$ by the weighting factors $\alpha_t^{g(i+1)}$ and $\alpha_t^{r(i+1)}$ separately. This weighted combination method is used to adjust the importance and contribution of the two multimodal features to the generation of descriptions, enabling the model to better utilize information from different features.

Following this, to further model the deep relationship between language context and multimodal features, we add the result of the weighted sum of multimodal features to the word feature $w_t^{i+1}$ and then normalize the result to obtain the final comprehensive feature representation $p_t^{i+1}$ with enhanced expressiveness and adaptability. The following are the matching definitions:

$$\alpha_t^{g(i+1)} = \sigma(W'^g[g_t^{R(i+1)}; w_t^{i+1}] + b'^g), \tag{25}$$

$$\alpha_t^{r(i+1)} = \sigma(W'^r[r_t^{G(i+1)}; w_t^{i+1}] + b'^g), \tag{26}$$

$$p_t^{i+1} = LayerNorm(g_t^{R(i+1)} \otimes \alpha_t^{g(i+1)} + r_t^{G(i+1)} \otimes \alpha_t^{r(i+1)} + w_t^{i+1}). \tag{27}$$

The multimodal comprehensive feature $p_t^{i+1}$ expands the capability for complex cross-modal reasoning, and it will be fed into subsequent feedforward neural networks and decoding layers for further decoding.

## 3.3 Training Details

As is customary in image captioning research[6, 32], we use cross-entropy loss (XE) for pre-training the model and reinforcement learning for fine-tuning. Specifically, during the XE training phase, with the target true sequence $w_{1:T}^*$ provided, the goal is to reduce the cross-entropy loss (XE) defined as follows.:

$$L_{XE}(\theta) = -\sum_{t=1}^{T} \log(p_\theta(w_t^*|w_{1:t-1}^*)), \tag{28}$$

where $\theta$ is our model's parameter.

Next, we use the self-critical sequence training (SCST)[32] approach to constantly optimize the non-differentiable CIDEr-D score during the reinforcement learning phase as follows[6]:

$$\nabla_\theta L_{RL}(\theta) = -\frac{1}{k} \sum_{i=1}^{k} ((r(w_{1:T}^i) - b)\nabla_\theta \log p_\theta(w_{1:T}^i)), \tag{29}$$

where $k$ is the beam search size, $r$ is the CIDEr-D score function, and $b = (\sum_i r(w_{1:T}^i))/k$ is the baseline.

## 4 EXPERIMENTS

### 4.1 Experimental Settings

**Dataset.** We conducted experiments on the MSCOCO 2014 dataset[20], which consists of 123,287 images, including 82,783 training images, 40,504 validation images, and 40,775 test images, each with 5 different annotations. To ensure a fair comparison with most existing techniques, we utilized the splits provided by Karpathy et al.[12], where 5,000 images were used for validation, 5,000 images were used for testing. The remaining images were used for training. Additionally, MSCOCO provides 40,775 images for online evaluation, with their annotations not publicly available. During the training process, we converted all training captions to lowercase

and removed words that appeared less than 5 times, resulting in a vocabulary of 10,201 words.

**Evaluation Metrics.** The effectiveness of image descriptions is evaluated using a variety of captioning metrics, encompassing BLEU (B@1-4)[28], METEOR (M)[3], ROUGE (R)[5], CIDEr (C)[38], and SPICE (S)[1].

**Implementation Details.** In DCCT, we used the Faster R-CNN object detector provided by VinVL[50] and the pre-trained CLIP-RN50×4[31] model to extract region features and grid features, respectively. The grid size was 9 by 9, and the maximum number of items in region features was 50. The dimensionality of the grid features was 2560, while the dimensionality of the extracted region features was 2048. We set the number of cascade layers to 1, the number of probability sparse attention layers in the encoder to 3, and the number of layers in the decoder to 3. In addition, we set the hyperparameters for DCCT training by implementing the Transformer model as suggested in[6]. The feed-forward neural network (FFN) layer had an inner dimension of 2048, the multi-head attention mechanism was configured with 8 heads, the dimensionality $d$ of each layer was set to 512, and the dropout after each multi-head attention and FFN layer was set to 0.1. During the cross-entropy training stage, we used the Adam optimizer to train our model until the CIDEr metric continuously decreased for five epochs. At that point, we switched to self-critical sequence training, i.e., reinforcement learning stage. The batch size was set to 20. Additionally, the learning rate scheduling approach was implemented using the following[51]:

$$lr = \begin{cases} base\_lr * e/4, & e \le 3, \\ base\_lr, & 3 < e \le 10, \\ base\_lr * 0.2, & 10 < e \le 12, \\ base\_lr * 0.2 * 0.2, & otherwise, \end{cases} \quad (30)$$

where $e$ is the number of iterations that are currently being performed, starting at 0, and the base learning rate ($base\_lr$) was set to 0.0001. During the reinforcement learning phase, we optimized the model employing the Adam optimizer with a batch size of 100 and a fixed learning rate of $5 \times 10^{-6}$. The training process was stopped after five epochs of progressively decreasing CIDEr metrics. We employed a beam search approach with a beam size of five during inference.

## 4.2 Ablation Study

To show the efficacy of the suggested distillation cascade fusion encoder (DCFE) and parallel cross-fusion attention module (PCFA) and their effects on the overall performance of DCCT on the MS COCO dataset, we carried out some ablation experiments in this section. The outcomes of our ablation trials are displayed in Table 1.

As can be seen in Table 1, the Transformer-based Base model performs worse overall with a CIDEr score of only about 122.9 when the dual visual features are not fused, as demonstrated in the second and third rows of the table, i.e., when either grid features or region features are used for image captioning. In the cross-attention module, we usually fuse both region and grid features when they are used for captioning images.

We initially fixed the encoder section as the basic Transformer encoder module "Base". Then we compared the overall performance

**Table 1: The ablation experiment results of DCCT on the COCO Karpathy split are as follows. "W/o Fusion" refers to the model not using feature fusion and only utilizing grid or region features. "W/ Fusion" indicates the model using both grid and region features and performing feature fusion during the decoding phase. "Base" represents the baseline Transformer-based encoder-decoder structure model.**

| | Encoder | Cross Attention | B@4 | M | R | C | S |
|---|---|---|---|---|---|---|---|
| W/o Fusion | Base(Grid) | Base(Grid) | 38.0 | 29.1 | 57.9 | 122.6 | 21.8 |
| | Base(Region) | Base(Region) | 38.0 | 29.2 | 57.9 | 122.9 | 21.8 |
| W/ Fusion | Base | Cascade Fusion | 38.2 | 29.1 | 58.1 | 123.9 | 21.9 |
| | Base | Sequential Fusion | 38.3 | 29.2 | 58.4 | 124.6 | 21.9 |
| | Base | PCFA | 38.6 | 29.4 | 58.4 | 125.6 | 22.1 |
| | DCFE | Cascade Fusion | 38.5 | 29.6 | 58.6 | 124.8 | 21.5 |
| | DCFE | Sequential Fusion | 39.0 | 29.5 | 58.8 | 125.1 | 22.6 |
| | DCFE | PCFA | **39.5** | **30.2** | **58.9** | **127.5** | **23.3** |

of various cross-attention modules with our suggested PCFA to confirm the efficacy of our proposed PCFA. This is shown in rows 4, 5, and 6 of Table 1. "Concat Fusion" denotes the cascade fusion of dual visual features, which are subsequently supplied into the decoder to decode. "Sequential Fusion" represents sequential fusion, where cross-attention is separately computed for region and grid features, and their attention results are added together for subsequent decoding. As indicated by the experimental results in the table, the model's overall performance is higher after utilizing feature fusion techniques in the cross-attention module than when using only one feature type. Furthermore, when using only cascade fusion, the CIDEr score is 1.3 points higher than when using only grid features. However, whether using cascade fusion or sequential fusion, the performance improvement is minimal, with a difference of only 0.7 between the two. Nevertheless, when we replace cascade fusion and sequential fusion with PCFA, the CIDEr score increases by 1.7 compared to cascade fusion and by 1.0 compared to sequential fusion. This indicates that our PCFA can fully utilize the correlation and complementarity between grid and region features, thereby achieving better characteristic fusion and expressive capability.

To analyze the effectiveness of DCFE in the encoding phase, we kept the cross-attention part of the decoder fixed, using cascade fusion, sequential fusion, and PCFA, as indicated in rows 4 and 7 of Table 1. When the cross-attention part of the decoder was fixed as cascade fusion, our encoder DCFE significantly improved the model's overall performance in comparison with the Transformer-based encoder, with the CIDEr score increasing by approximately 1.0. This is because our encoder, during feature encoding, filters out some redundant features through the probabilistic sparse attention layer, allowing attention to focus more on important feature parts and making subsequent decoding simpler and more efficient. Subsequently, when the cross-attention module was fixed as sequential fusion, there was also an improvement in model performance. We consider the model architecture in row 4 as our baseline model. When we equipped our DCFE and PCFA on the baseline model, the CIDEr score achieved the best performance at 127.5, further demonstrating the effectiveness of our proposed encoder and cross-attention module for image caption generation.

**Table 2: The effectiveness of different approaches on the MSCOCO Karpathy test split(single model configuration).**

| | Cross-Entropy Loss | | | | | | CIDEr Score Optimization | | | | | |
|---|---|---|---|---|---|---|---|---|---|---|---|---|
| | B@1 | B@4 | M | R | C | S | B@1 | B@4 | M | R | C | S |
| Up-Down[2] | 77.2 | 36.2 | 27.0 | 56.4 | 113.5 | 20.3 | 79.8 | 36.3 | 27.7 | 56.9 | 120.1 | 21.4 |
| SGAE[47] | 77.3 | 36.8 | 27.9 | 57.0 | 116.3 | 20.9 | 81.0 | 39.0 | 28.4 | 58.9 | 129.1 | 22.2 |
| AoANet[11] | 77.4 | 37.2 | 28.4 | 57.5 | 119.8 | 21.3 | 80.2 | 38.9 | 29.2 | 58.8 | 129.8 | 22.4 |
| X-Transformer[27] | 77.3 | 37.0 | 28.7 | 57.5 | 120.0 | 21.8 | 80.9 | 39.7 | 29.5 | 59.1 | 132.8 | 23.4 |
| $M^2$ Transformer[6] | - | - | - | - | - | - | 80.8 | 39.1 | 29.2 | 58.6 | 131.2 | 22.6 |
| RSTNet[51] | - | - | - | - | - | - | 81.8 | 40.1 | 29.8 | 59.5 | 135.6 | 23.3 |
| DLCT[23] | - | - | - | - | - | - | 81.4 | 39.8 | 29.5 | 59.1 | 133.8 | 23.0 |
| Dual Global[44] | - | - | - | - | - | - | 81.3 | 40.3 | 29.2 | 59.4 | 132.4 | 23.3 |
| DIFNet[43] | - | - | - | - | - | - | 81.7 | 40.0 | 29.7 | 59.4 | 136.2 | 23.2 |
| VinVL[50] | - | 38.2 | 30.3 | - | 129.3 | 23.6 | - | 40.9 | 30.9 | - | 140.4 | 25.1 |
| COS-Net[17] | 79.2 | 39.2 | 29.7 | 58.9 | 127.4 | 22.7 | 82.7 | 42.0 | 30.6 | 60.6 | 141.1 | 24.6 |
| DLRN[40] | - | - | - | - | - | - | 81.1 | 38.6 | 28.5 | 58.8 | 128.9 | 22.1 |
| TLG[30] | - | - | - | - | - | - | 86.1 | 37.8 | 39.2 | 65.1 | 132.9 | - |
| LSTNet[24] | - | - | - | - | - | - | 81.5 | 40.3 | 29.6 | 59.4 | 134.8 | 23.1 |
| HAAV[13] | - | - | - | - | - | - | - | 41.0 | 30.2 | - | 141.5 | 23.9 |
| **DCCT** | **79.0** | **39.5** | **30.2** | **58.9** | **127.5** | **23.3** | **83.2** | **42.7** | **30.6** | **60.8** | **141.7** | **24.6** |

**Table 3: The performances of various methods on COCO Karpathy test split (ensemble model setup).**

| | Cross-Entropy Loss | | | | | | CIDEr Score Optimization | | | | | |
|---|---|---|---|---|---|---|---|---|---|---|---|---|
| | B@1 | B@4 | M | R | C | S | B@1 | B@4 | M | R | C | S |
| AoANet[11] | 80.2 | 38.9 | 29.2 | 58.8 | 129.8 | 22.4 | 81.6 | 39 | 28.4 | 58.9 | 129.1 | 22.2 |
| M2 Transformer[6] | - | - | - | - | - | - | 82.0 | 40.5 | 29.7 | 59.5 | 134.5 | 23.5 |
| X-Transformer[27] | 77.8 | 37.7 | 29.0 | 58.0 | 122.1 | 21.9 | 81.7 | 40.7 | 29.9 | 59.7 | 135.3 | 23.8 |
| DLCT[23] | - | - | - | - | - | - | 82.2 | 40.8 | 29.9 | 59.8 | 137.5 | 23.3 |
| COS-Net[17] | 79.6 | 40.0 | 30.0 | 59.4 | 129.5 | 22.9 | 83.5 | 42.9 | 30.8 | 61.0 | 143.0 | 24.7 |
| **DCCT** | **79.7** | **40.7** | **30.0** | **59.5** | **130.5** | **22.8** | **84.2** | **43.4** | **31.2** | **61.2** | **144.1** | **25.0** |

## 4.3 Comparisons with State-of-the-Art

Using two distinct dataset splits—the official online evaluation test set split and the standard Karpathy test set split—we evaluated our DCCT with several advanced image description techniques. We assessed the effectiveness of ensemble models as well as single models for the Karpathy test set split.

**Offline Evaluation.** We assess our technique DCCT's performance on the MSCOCO Karpathy test set against the state-of-the-art model at this time. Table 2 is an illustration of the performance of a single model during two different training periods. Furthermore, we show the ensemble model's outcomes for a thorough comparison. Consistently outperforming the others in all metrics, our single model can be observed to show superior performance. Firstly, compared to traditional image captioning methods such as Up-Down[2], $M^2$[6], X-Transformer[27], and DLCT[23], our CIDEr score is on average improved by around 10 points. This improvement is attributed to the powerful visual features provided by CLIP[31] and VinVL[50], as well as our proposed distillation cascade fusion encoder. Although most of our metrics surpass the existing advanced methods, our SPICE metric remains comparable. Secondly, compared to RSTNet[51] and DIFNet[43], our CIDEr

scores are improved by 6.1 and 5.5, respectively. We also compare DCCT with the popular large-scale vision language model VinVL, which uses a large-scale image-text dataset to pre-train image captioning models. In contrast, our method is trained from the ground up, with no prior instruction. The efficacy of our proposed DCCT in the domain of image captioning is demonstrated by the 1.3-point enhancement of CIDEr scoring that our method maintains, as shown in Table 2. For the cross-modal retrieval, the grid features are additionally extracted by the COS-Net model with the help of the pre-trained CLIP model. Compared to this model, our CIDEr score is still improved by 0.6, and comparable performance is achieved in other metrics. We also contrasted our model with more sophisticated models, like HAAV[13], TLG[30], and LSTNet[24]. According to the bleu-1 metric, our model trails TLG by 3.1 points, indicating a minor lag. We beat LSTNet by 6.9 points, but we performed better by about 8.8 points in the CIDEr metric. Further demonstrating the advantages of our suggested DCCT, we also show a certain level of competitiveness with HAAV, with comparable performance.

**Ensemble Model.** We conducted an ensemble evaluation using four independently trained models with distinct random seeds. As depicted in Table 3, it presents the performance of various ensemble

**Table 4: Leaderboard on the COCO online test server for different methods.**

| | BLEU-1 | | BLEU-2 | | BLEU-3 | | BLEU-4 | | METEOR | | ROUGE | | CIDEr | |
|---|---|---|---|---|---|---|---|---|---|---|---|---|---|---|
| | c5 | c40 | c5 | c40 | c5 | c40 | c5 | c40 | c5 | c40 | c5 | c40 | c5 | c40 |
| Up-Down[2] | 80.2 | 95.2 | 64.1 | 88.8 | 49.1 | 79.4 | 36.9 | 68.5 | 27.6 | 36.7 | 57.1 | 72.4 | 117.9 | 120.5 |
| AoANet[11] | 81.0 | 95.0 | 65.8 | 89.6 | 51.4 | 81.3 | 39.4 | 71.2 | 29.1 | 38.5 | 58.9 | 74.5 | 126.9 | 129.6 |
| SGAE[47] | 81.0 | 95.3 | 65.6 | 89.5 | 50.7 | 80.4 | 38.5 | 69.7 | 28.2 | 37.2 | 58.6 | 73.6 | 123.8 | 126.5 |
| X-Transformer[27] | 81.9 | 95.7 | 66.9 | 90.5 | 52.4 | 82.5 | 40.3 | 72.4 | 29.6 | 39.2 | 59.5 | 75.0 | 131.1 | 133.5 |
| $M^2$ Transformer[6] | 81.6 | 96.0 | 66.4 | 90.8 | 51.8 | 82.7 | 39.7 | 72.8 | 29.4 | 39.0 | 59.2 | 74.8 | 129.3 | 132.1 |
| DLCT[23] | 82.4 | 96.6 | 67.4 | 91.7 | 52.8 | 83.8 | 40.6 | 74.0 | 29.8 | 39.6 | 59.8 | 75.3 | 133.3 | 135.4 |
| RSTNet[51] | 82.1 | 96.4 | 67.0 | 91.3 | 52.2 | 83.0 | 40.0 | 73.1 | 29.6 | 39.1 | 59.5 | 74.6 | 131.9 | 134.0 |
| COS-Net[17] | 83.3 | 96.8 | 68.6 | 92.3 | 54.2 | 84.5 | 42.0 | 74.7 | 30.4 | 40.1 | 60.6 | 76.4 | 136.7 | 138.3 |
| LSTNet[24] | 82.6 | 96.7 | 67.8 | 92.0 | 53.3 | 84.3 | 41.1 | 74.7 | 29.9 | 39.6 | 60.0 | 75.4 | 134.0 | 136.3 |
| TDANet[29] | 83.8 | 97.1 | 64.2 | 88.3 | 53.3 | 83.7 | 37.8 | 70.5 | 35.6 | 42.4 | 61.1 | 78.2 | 132.0 | 132.6 |
| **DCCT** | **83.8** | **97.4** | **69.5** | **93.1** | **54.9** | **86.2** | **42.7** | **76.4** | **30.7** | **40.6** | **61.2** | **77.1** | **138.1** | **140.2** |

models during two distinct training stages. It can be observed that in the ensemble setting, our proposed DCCT model achieves a CIDEr score of 144.1, outperforming all current methods. In particular, we show that our suggested DCCT is superior to COS-Net by 1.1 points on the CIDEr metric in the image description task.

**Online Evaluation.** To better confirm the effectiveness of our image captioning model DCCT, we submitted it to the MS COCO online test server for evaluation and compared its performance with other online models. For the online evaluation, since most top-performing methods on the leaderboard use ensemble models, we also used the aforementioned ensemble configuration for a fair comparison. In Table 4, we respectively recorded the overall effectiveness of the model employing five reference descriptions(c5) along with forty standard descriptions(c40). Observing that our approach surpasses the current advanced methods across all metrics, it attains the highest performance level. Compared to COS-Net, our model achieves scores of 76.4 and 140.2 in BLEU-4 (c40) and CIDEr (c40) respectively, which are 1.7 and 1.9 points higher than the best-performing methods.

## 4.4 Qualitative Analysis

In this section, we conducted a detailed qualitative analysis of the proposed DCCT model, with Figure 4 showing examples of descriptive sentences generated by DCCT and the Transformer baseline, along with basic factual sentences annotated by humans (GT). It can be observed that both DCCT and the Transformer baseline models are capable of generating coherent language descriptions. However, when redundant feature information is filtered out, some finer-grained information becomes more prominent, making it easier for our DCCT model to capture these refined details and generate more accurate descriptions. For example, in the first instance, due to the inability of the baseline model to filter out certain redundant features and capture finer-grained information, an incorrect key descriptor "balloon" is generated, whereas we produce "kite," which matches the ground truth annotation. Simultaneously, to mitigate the negative effects of regional features, DCCT combines grid features to compensate for them, resulting in more accurate positioning and fine-grained content. For example, in the first instance, the model generates "next to" and in the fourth example it generates "broccoli".

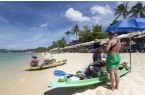
DCCT: A couple standing on the beach next to their boat.
Transformer: some people hanging out on a beach.
GT1: A man is getting food from a beach side boat.
GT2: A fat and bald shirtless man stands next to a boat.

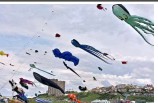
DCCT: The sky is filled with many colorful flying kites.
Transformer: A large number of different balloons in the air.
GT1: A bunch of colorful kites flying in the sky.
GT2: The sky is full of many colorful flying kites.

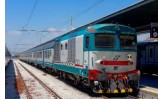
DCCT: A train on the tracks parked at a train station.
Transformer: The train is posted at a train station.
GT1: A train on the tracks with a little bit of graffiti.
GT2: A passenger train pulling into a station platform.

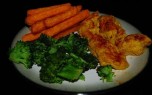
DCCT: A dinner plate with fried ham, eggs, and broccoli.
Transformer: A plate of fried food, egg, and vegetables.
GT1: A dinner plate with vegetables and meat on it.
GT2: A dinner plate with some fried food, eggs, and broccoli.

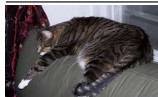
DCCT: A striped cat sleeping on the back of grey couch.
Transformer: A black cat sleeping on the sofa.
GT1: A sleeping cat laying on the backside of a sofa.
GT2: A striped cat sleeping on the back of a gray leather couch.

**Figure 4: Qualitative results of our DCCT and Transformer, coupled with ground-truth descriptions**

## 5 CONCLUSION

In this paper, we propose a novel Distilled Cross-Combination Transformer(DCCT) image captioning model, which achieves the cross-fusion of refined visual features with textual features. In DCCT, we provide a distillation cascade fusion encoder that improves regional and grid feature extraction by eliminating superfluous features from images and providing the decoder with more precise visual data. To further achieve a more comprehensive multimodal feature representation, we also offer a parallel cross-fusion attention module that fully utilizes the complementarity and correlation between the dual visual characteristics via gating, cross-attention, and parallel computing. Comprehensive studies on the MSCOCO dataset demonstrate the suggested DCCT strategy realizes remarkable effectiveness and exceeds numerous state-of-the-art approaches.

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
