# OpenReview forum: "Distilled Cross-Combination Transformer for Image Captioning with Dual Refined Visual Features"
_acmmm.org/ACMMM/2024/Conference — MM2024 Poster_

### Official Review · Reviewer_Zsro · 2024-05-23

**Rating:** 4
**Confidence:** 4

**Summary:**

The paper presents the Distilled Cross-Combination Transformer (DCCT) for image captioning, which uses both region and grid visual features. It introduces a Distillation Cascade Fusion Encoder (DCFE) to refine visual features by filtering out redundancies and a Parallel Cross-Fusion Attention (PCFA) module to combine the features effectively. Experiments on the MSCOCO dataset show that DCCT performs well across various metrics compared to existing methods. The approach addresses the integration of multiple types of visual features and their fusion in generating image captions. The complexity of the model and its validation on other datasets are mentioned as considerations for further research.

**Strengths:**

-The figure are very clear and easy to understand.
-The writing is straightforward and easy to follow.
-The model achieves high performance across various metrics on the MSCOCO dataset.

**Limitations:**

-PCFA is a lot like [a]; you need to talk more about the differences, like how the features interact.
-Show some actual numbers to prove that DCFE speeds up inference after filtering tokens.
-Include visualizations of DCFE's effects so we can see the results clearly.
-The performance improvement is pretty minor, and the bolding in the first column of Table 2 looks wrong.
-You missed discussing several important works like [b]; you should compare them in the related work and in Tables 2, 3, and 4.

If the authors address these issues, I will consider raising the score.

[a] Dual-Level Collaborative Transformer for Image Captioning. AAAI 2021.
[b] Improving Image Captioning by Leveraging Intra- and Inter-layer Global Representation in Transformer Network. AAAI 2021.

**Suitability:**

3

---

### Official Review · Reviewer_BbCu · 2024-05-23

**Rating:** 2
**Confidence:** 2

**Summary:**

This paper focuses on the effectiveness and efficiency of the image caption, where an architecture named DCCT is proposed. Specifically, DCCT contains the DCFE and PCFA, the former is designed to filter out redundant visual features based on the attention weights, and the PCFA is proposed to fuse the multi-modal features better.

**Strengths:**

1. The motivation of the paper is clear and the proposed modules are effective.

**Limitations:**

1. The novelty of the proposed method is limited, for example [1] has already proposed a gated-attention mechanism to select a subset of elements to attend. The PCFA also seems trivial with only interactions across multi-modal features;
2. The performance of the proposed method is not the best in METEOR, the bold is not appropriate;
3. It is better to provide  a performance of CLIP as a baseline, since the proposed method adopts the CLIP as a feature extractor.


[1] Not All Attention Is Needed: Gated Attention Network for Sequence Data, AAAI, 2020

**Suitability:**

2

---

### Official Review · Reviewer_QZDs · 2024-05-24

**Rating:** 5
**Confidence:** 3

**Summary:**

This paper proposes a novel Transformer-based image captioning framework. The proposed method compresses tokens with less impact to focus the attention on important features. The proposed method combines grid features and region features, achieving advanced performance on MSCOCO dataset.

**Strengths:**

1. This paper introduces distillation into the image captioning task, and the proposed method compresses tokens with less impact to focus the attention of the model on important features. This method demonstrates high innovation.
2.The proposed method utilizes cross-attention to integrate grid features and region features. The simplicity and effectiveness of this method have also been demonstrated through comparison studies.
3. Sufficient experiments have been conducted on the MSCOCO dataset to demonstrate the effectiveness of this method.

**Limitations:**

1.The Motivation expression reflected in Figure 1 (b) is slightly inferior. The rationality of the two proposed questions needs to be verified and supported by relevant papers. Additionally, there is ambiguity between Figure 1 (b) and the description of the problem.
(i)The first type of understanding is that according to Figure 1 (b), both G and R of Sequential Fusion are cross-attention to each other's V, Q, and K. For the first question, since both G and R of Sequential Fusion have cross-attention as V, Q, and K respectively, why is it that, as stated in the paper, "the interaction between different features cannot be fully utilized"? For the second question, although the order will affect the final performance, the model still considers both orders, so there is still a lack of argumentation and persuasiveness.
(ii)The second type of understanding, according to the textual expression in the paper, is that both G and R in Sequential Fusion are self-attention rather than cross-attention in Figure 1 (b). So, for the second question, since the Transformer block uses a position encoding sequence and there is no forgetting problem as in LSTM, there will be no issue of insufficient attention on the other due to the flipping of the order.
2. The paper lacks proof for the first motivation about distillation. We’d like to investigate the changes in network attention before and after distillation in order to verify the effectiveness of the first motivation.
3. The paper lacks visualization results for word generation to prove the second motivation. This type of visualization results are important as it demonstrates the grid features and region features that each generated word focuses on.

**Suitability:**

3

---

### Meta-Review · Area_Chair_ghVS · 2024-07-03

**Recommendation:** Accept (Poster)
**Confidence:** 5

**Metareview:**

The paper presents the Distilled Cross-Combination Transformer (DCCT) for image captioning, which uses region and grid visual features. It introduces a Distillation Cascade Fusion Encoder (DCFE) to refine visual features by filtering out redundancies and a Parallel Cross-Fusion Attention (PCFA) module to combine the features effectively. The related experiments have demonstrated the effectiveness of the method. Some reviewers expressed concerns about the paper's novelty, but after one rebuttal round, the authors provided convincing responses, and the reviewers acknowledged its contribution to the image captioning field. Hopefully, the authors will further improve the paper based on the reviewers' feedback.